# Plastic in the Environment: A Modern Type of Abiotic Stress for Plant Physiology

**DOI:** 10.3390/plants12213717

**Published:** 2023-10-29

**Authors:** Giorgia Santini, Daniela Castiglia, Maryanna Martina Perrotta, Simone Landi, Giulia Maisto, Sergio Esposito

**Affiliations:** 1Department of Biology, University of Naples “Federico II”, Via Cinthia, I-80126 Napoli, Italy; giorgia.santini@unina.it (G.S.); maryannamartina.perrotta@unina.it (M.M.P.); giulia.maisto@unina.it (G.M.); sergio.esposito@unina.it (S.E.); 2Bio-Organic Chemistry Unit, Institute of Biomolecular Chemistry CNR, Via Campi Flegrei 34, Pozzuoli, 80078 Naples, Italy

**Keywords:** agriculture, microplastic, nanoplastic, oxidative, soil, transcriptomic, trace metals

## Abstract

In recent years, plastic pollution has become a growing environmental concern: more than 350 million tons of plastic material are produced annually. Although many efforts have been made to recycle waste, a significant proportion of these plastics contaminate and accumulate in the environment. A central point in plastic pollution is demonstrated by the evidence that plastic objects gradually and continuously split up into smaller pieces, thus producing subtle and invisible pollution caused by microplastics (MP) and nanoplastics (NP). The small dimensions of these particles allow for the diffusion of these contaminants in farmlands, forest, freshwater, and oceans worldwide, posing serious menaces to human, animal, and plant health. The uptake of MPs and NPs into plant cells seriously affects plant growth, development, and photosynthesis, finally limiting crop yields and endangering natural environmental biodiversity. Furthermore, nano- and microplastics—once adsorbed by plants—can easily enter the food chain, being highly toxic to animals and humans. This review addresses the impacts of MP and NP particles on plants in the terrestrial environment. In particular, we provide an overview here of the detrimental effects of photosynthetic injuries, oxidative stress, ROS production, and protein damage triggered by MN and NP in higher plants and, more specifically, in crops. The possible damage at the physiological and environmental levels is discussed.

## 1. Introduction

In early 1900s, the innovation brought by the invention of plastics drastically revolutionized the world; in particular, after the end of World War II, plastics production boomed when military factories manufacturing plastic for war purposes revised their productions to civil use [1]. Currently, plastics are utilized for a wide range of items, from food packaging to technological applications, due to their resistance to corrosion, low density, and low electric and thermal conductivities [2]. In particular, one of the most significant advantages of the use of plastic material is the prolonged and complex degradation process, requiring from 250 to 1000 years. This aspect has become a significant environmental drawback, considering that, depending on their chemical structure, plastics decay over a long period and accumulate globally [3]. Since the early 1990s the dramatic implications of plastics for the environment have been proven, thus raising the concern of international institutions. In recent years, the presence of large quantities of microplastics (MPs) and nanoplastics (NPs) has been observed in the environment due to the degradation and dissemination of plastic waste [4].

More specifically, MPs are plastic particles with sizes from than 1 μm to 5 mm [1]. These particles include fragments smaller than 1 μm, which are considered NPs [4]. MPs are divided into primary MPs, from chemically derived products used in pharmaceutical industries, and secondary MPs, formed when large plastics are broken down through physical, chemical, and biological processes [5,6]. MPs and NPs can exist as fragments, microfibers, spheres, films, and pellets [1].

Of course, both MPs and NPs can be classified based on the plastic polymers from which they originate, so there are MPs and NPs formed of polyamide (PA), polyethylene (PE), polyethylene terephthalate (PET), polyvinyl chloride (PVC), and polybutylene adipate terephthalate (PBAT). PE and PET are mainly used for packaging, bottles, and bags, while PVC is used for industry and construction [7]. PE and PET produce the most easily transportable MPs, especially through wind erosion, considering their low density [8].

Currently, MP and NP pollution is considered one of the most problematic environment-related issues, being widespread in deserts, tropics, deep oceans, and the Arctic and Antarctic poles [2]. Only considering aquatic environments, an annual release of 19–23 million tons of plastic residues has been estimated [1]. In the terrestrial environment, the release of plastic pollution, especially caused by poorly managed waste-disposal practices, is gaining increasing attention due the impact of plastic particles on plant growth, and animal and human health.

The aim of this work is to review the current knowledge and findings about the occurrences, effects, accumulation, and influences of MPs and NPs on different plant organisms. In particular, the detrimental effects triggered by MPs and NPs are summarized, including nutrient uptake, photosynthetic injuries, oxidative stress, ROS production, transcriptomic modification, and protein damage. The relationships between plastics and metals in the soil, how MPs can act as carriers of these pollutants, and the consequences for plant physiology are discussed.

## 2. Origin and Implications of Plastics in the Agricultural Environment

Soil is a heterogeneous habitat formed by a complex assemblage of minerals and organic matter, comprising a complex network of spaces filled with water and air [9]. MPs and NPs affect terrestrial environments, considering that these particles are solid contaminants similar in size and shape to those of the composing soil [10]. It has been observed that the presence of MPs and NPs can alter different physio-chemical properties of soil, such as bulk density, water retention capacity, pH, and organic matter (OM) content [10]. These aspects are critical for the soil system, impacting plant growth, development, and physiology. It is worth pointing out that soil properties are not only directly influenced by the presence of MPs and NPs, but they are also responsible for the bioavailability of the plastic particles influencing the activities of different protein receptors of microorganisms, animals, and plants [11].

MPs and NPs may also contain toxic compounds including pesticides, phthalates, endocrine-disrupting chemicals (EDCs), polycyclic aromatic hydrocarbons (PAHs), and trace metals. Therefore, MPs and NPs can act as vectors, transporting these compounds to the soil and causing them to become available depending on the temperature, ultraviolet radiation, pH, and oxygen content [11]. The presence of these toxic molecules in MPs and NPs negatively affects root growth, as well as disturbing the interaction between plants and their symbionts, damaging plant physiology and metabolism [12].

The utilization of plastics in agriculture has risen in recent years, with the specific aim of enhancing crop productivity and reducing food loss [13]. Plastics are commonly used for the construction of greenhouses and tunnels and to cover soil with mulch, shade cloth, pesticide containers, protective mesh, and irrigation tubing. Different estimations of the use of plastics in agriculture have been reported by several authors (Table 1).

One of the most widespread applications of plastic in agriculture consists of mulches mainly made of PE, PVC, and polypropylene (PP). Mulches effectively suppress weed growth, prevent soil erosion, and increase the soil temperature, leading to improved crop quality and yield [17]. Nevertheless, the potential advantages of plastic mulches and their widespread and prolonged use—coupled with a lack of proper management—result in the accumulation of plastic debris (MPs and NPs) in the soil. In the agricultural field, the removal of plastic films from soil at the end of the growing season is a labor-intensive process; moreover, it is impractical and cost-prohibitive to completely extract all the small fragments of the mulch film [18]. These residual plastic films result in the accumulation of MPs and NPs in the agroecosystems, increasing the effects of these toxic pollutants on plants and on the agricultural food chain [19].

In recent years, eco-friendly biodegradable plastics have been introduced. Their structural and surface characteristics allow for the attack of enzymes from the soil microbiota and biodeterioration, substantially reducing plastic pollution [20]. On the other hand, similar to conventional plastics, fragmentation of bioplastic mulches can occur by a range of environmental factors, thus releasing micro-bioplastics into agricultural soils [21].

## 3. Plastics as Abiotic Stress: General Effects on Plant Physiology and Metabolism

Is it possible to consider plastic contamination an abiotic stress for plants? Plants are not supposed to uptake or transfer MPs due their high molecular weight and large size; however, recent studies have highlighted that the presence of MPs can differentially affect plant growth, development, and physiology [4,22]. On the other hand, different crops have shown the ability to assimilate NPs penetrating into plant cell walls [23,24].

The effective impact of these particles on plant physiology is related to plastic quantities, cultivation strategies (e.g., soil or hydroponic), types, charges, and sizes of particles. In recent years, different authors have tested a range of MPs and NPs derived from PE, PP, PVC, and polystyrene (PS), showing critical effects on plants [24,25,26,27].

PS–MPs have shown detrimental effects on photosynthetic parameters, namely *Fv*/*Fm*, *Fv*/*P_0_*, and chlorophyll content, both in soil and in hydroponic cultivation systems in *Brassica chinensis* and *Lemna minor* L. [25,26]. Interestingly, *L. minor* L. plants showed highly reduced effects when cultivated with 100 mg L^−1^ of PS-MPs, while serious disadvantages were noted when they were cultivated with 200 mg L^−1^ [26].

Pignatelli et al. [27] analyzed the effects in soil of PE–, PVC–, and PP–MPs with dimensions ≤ 0.125 mm on *Lepidium sativum* L. The authors reported detrimental effects for each plastic type analyzed, indicating that PE treatment induced the greater increase in H_2_O_2_, and PVC treatment showed the highest proline content [27]. These results suggest the activation of different abiotic stress response pathways to oxidative and osmotic damages, respectively [28,29].

The comparison of the effects of 10-µm PS–MPs and PVC–MPs on hydroponic rice showed PVC–MPs to be the most damaging [30]. In particularly, 3 mg L^−1^ of PVC–MPs reduced root and shoot weight, net photosynthesis, and stomatal conductance; unexpectedly, the use of PVC–MPs also induced a significant and enormous increase in water use efficiency (WUE) [30].

Furthermore, MPs and NPs alter the soil structure and properties interfering with the uptake of water and nutrients by crops and reducing their yield [9,10]. As a consequence, plastic residues in the soil induced detrimental effects on root architecture, impacting the horizontal and vertical distributions of roots. For example, it was shown that root biomass and grain yield could be reduced by 30% and 9%, respectively, in *Zea mays* L. when the residual plastic films in soil reached 300 kg ha^−1^ [31].

Different studies have focused on the oxidative damages induced by MPs and NPs in higher plants, demonstrating that the toxic effects of MPs and NPs are dose- and size-dependent [32]. In particular, accumulation of NPs activates scavenging enzymes, such as catalase (CAT), superoxide dismutase (SOD), peroxidase (POD), and ascorbate peroxidase (APX), which are generally involved in the response to abiotic stress [33]. These enzymes showed different behaviors depending on the model plant species and the type of used plastics (Figure 1).

In *Vicia faba* L., exposure to NPs is correlated with alterations in CAT, SOD, and POD enzymatic activities to counteract damage induced by the increase in ROS levels [33]. Exposure to 100 mg/L of PS–NPs with sizes of about 100 nm for 48 h induced strong increases in SOD activity [33]. Similar results were obtained for POD [30]. In contrast, CAT activity showed a significant decrease [33], likely correlated with the variation in H_2_O_2_ levels produced by SOD. Specific exposure to 100-nm NPs leads to the accumulation of these particles in the root tissue, disrupting the nutrient import and probably causing both oxidative damage and genotoxic effects [33].

Similarly, tomato plants (*Solanum lycopersicum* L.) were tested for the toxic effects of different levels of PS–, PE–, and PP–MPs [34]. It was demonstrated that exposure to PE and PP (more than PS) affected seed germination, inducing oxidative stress in plants. The toxic effects of MPs in tomato plants are correlated with the alteration in SOD activity, with an initial increase and a subsequent decrease with the increase in MPs levels; POD showed a decrease in activity with any MPs dosage; and CAT played a major role in long-term response [34].

The treatment of *Oryza sativa* plants with plastic particles of higher dimensions also induced the activation of antioxidant enzymes to counteract the effects of ROS [30]. In particular, PS–MPs and PVC–MPs increased the activities of SOD, POD, and CAT using concentrations of about 0.5, 1.5, and 3 mg L^−1^. Comparing the two different types of MPs, the PVC–MPs caused a faster and greater accumulation of H_2_O_2_ and malondialdehyde levels [30].

Arikan et al. [26] evaluated the potential effects of PS–NP exposure on the antioxidant system in *L. minor* L., exposing it to NPs in the medium (100 mg L^−1^ and 200 mg L^−1^). These treatments increased the activity of SOD by 3-fold and 4.6-fold, respectively. Similarly, results were obtained for CAT and APX. Interestingly, the authors suggested that NADPH-oxidase (NOX)—a crucial enzyme responsible for O_2_^−^ formation in plants—may be not involved in the response to NPs in *L. minor* L. [26].

In *Allium cepa*, NPs caused chromosome aberrations induced by the alteration of mitotic phases. This effect is related to the accumulation of ROS [35,36]. In particular, a gradual and dose-dependent decrease in mitotic index was demonstrated after 24 h, 48 h, and 72 h of incubation with 400 mg L^−1^ of NPs. The cytotoxicity induced by NPs is correlated with the down-regulation of cdc2, a fundamental gene coding for cell cycle regulation. The down-regulation of this gene is correlated with the mito-depressive activity of NPs by G2/M transition in living cells [37].

## 4. MPs and NPs Induce Transcriptomics Reorganization and Modify Plant Metabolism

Next-generation techniques, namely transcriptomics, proteomics, metagenomics, and metabolomics, have made available a number of datasets useful for exploring the relationships among different metabolic pathways in different plant species. Considering the serious environmental risk caused by MPs and NPs, these tools have been extensively used to investigate the effects of MPs and NPs in crops (Figure 2).

*Oryza sativa* plants treated with PS–NPs and graphene oxide nanosheets (GONs) were analyzed using RNA-seq [38,39], revealing a drastic re-organization of the transcriptome [38,39]. In particular, when plants were exposed to GON, 392 differentially expressed genes (DEGs) were identified (179 up-regulated and 213 downregulated). Plants treated with PS–NPs showed 592 DEGs (287 up-regulated and 305 down-regulated) [38]. Among these DEGs, specific genes were related to oxidative response and secondary metabolism (Figure 2). Furthermore, unusual expression of genes involved in carbon metabolism and the jasmonic acid pathway was reported in rice seedlings, together with a significant reduction in starch content and a corresponding increase in glucose levels [38].

The transcriptomic approach was used to investigate how genes involved in root development are regulated in presence of NPs [38]. PS–NPs regulate critical genes, namely *Os*MADS25, *Os*NAAT1, *Os*OFP2, and *Os*IRO2. *Os*MADS25 genes encodes for a transcription factor regulating lateral root development, associated with the nitrate signaling pathway [38]. Interestingly, *Os*NAAT1 and *Os*IRO2 are genes involved in the response to iron (Fe) uptake and deficiency, respectively [40,41]. These genes respond differently to NP treatment: the former is up-regulated, while the latter is down-regulated [38]. *Os*NAAT1 encodes for a nicotianamine aminotransferase (NA) that catalyzes the amino transfer of NA to produce an intermediate in the biosynthetic pathway of mugineic acid family phytosiderophores (MAs) [40]. The gene *Os*IRO2 encodes for a basic helix–loop–helix (bHLH) transcription factor, and its expression is activated by an Fe deficiency [41]. On the other hand, the NPs down-regulated the *Os*OFP2 gene, which encodes for a transcription factor involved in lignin biosynthesis and hormone homeostasis [42].
Figure 2Scheme of the main GO (Gene Ontology) and KEGG (Kyoto Encyclopedia of Gene and Genome) categories enriched in *Oryza sativa* plants subjected to different types of plastics contamination [38,39,43].
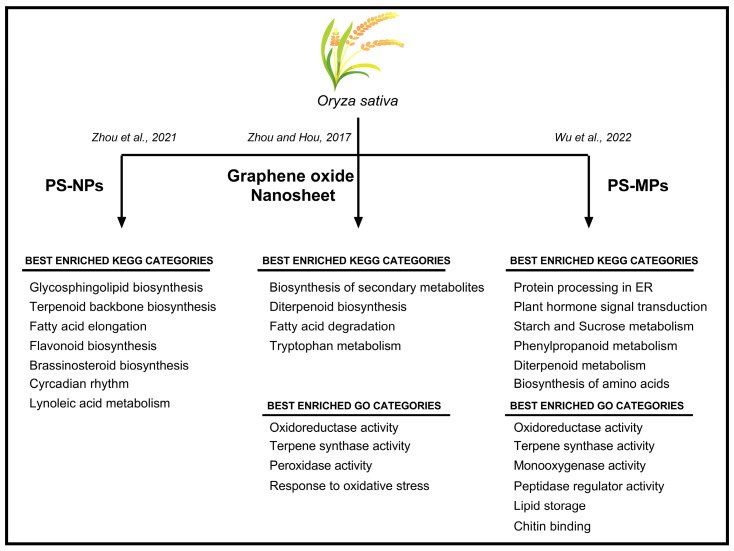


Furthermore, integration of metabolomics and transcriptomics was used to analyze the effects of PS–MPs in two different rice genotypes. Wu et al. [43] identified 22.597 DEGs in genotype Y900 and 23.151 in XS123 genotype. Interestingly, the authors indicated a genotype-specific response to treatments [43]. Both genotypes showed dose-dependent activation of different pathways to respond to PS–MP stress. Y900 showed the regulation of DEGs related to polysaccharide metabolism after exposure to a low dose of MPs and to lipid biosynthesis after exposure to a high dose of MPs. Instead, exposure to both the low and high doses of PS-MPs induced in the XS123 genotype changes in DEGs related to amino–sugar and nucleotide–sugar metabolism and hydrolase activity. The two genotypes also showed contrasting behaviors in enzymes related to the TCA cycle. ATP citrate synthase (ACLY), isocitrate dehydrogenase (IDH1), dihydrolipoyllysine-residue succinyl transferase (OADH), and succinate dehydrogenase (SDH3-2) were differently expressed. The TCA cycle was strongly activated in the XS123 genotype and inhibited in the Y900 genotype [43].

The exposure to PE–MPs of *Zea mays* L. and *Nicotiana tabacum* L. [44,45] showed interesting results. PE–MPs of different dimensions (1.08 g cm^−3^ and 1.35 g cm^−3^) on *Z. mays* L. roots affected 858 DEGs (624 up- and 234 down-regulated) and 1769 DEGs (1298 up- and 471 down-regulated), respectively. Transcriptomic analysis highlighted the activation of genes involved in the transport of different substrates as amino acids (similarly under different-sized MPs) and different sugars, namely galactose, glucose, and xylose, in a size-specific manner [44]. PE–MPs also affected the leaves of *N. tabacum* L. seedlings, influencing the expression of 5355 DEGs in plants grown in the presence of MPs (1000 g kg^−1^ in soil) [45]. The authors focused on photosynthesis, identifying that PE–MP stress significantly enriches different GO categories, particularly “photosynthesis, light harvesting in photosystem I”, “photosynthesis, light harvesting”, and “photosystem I reaction center”.

An alternative approach to investigation of the effects of MPs and NPs is represented by metagenomics due to the impact of plastics in terrestrial environments on microbial communities and their relationships with plants [46]. However, comprehensive studies of the impacts of residual plastics have been scarce; in particular, in-depth studies on soil-rhizosphere microbe-plant interaction remain elusive. The importance of metagenomics studies should be based on the ubiquitous interactions among microorganisms, plants, and animals, while plastics and related compounds can produce multiple impacts on the environmental microbiomes [47]. These aspects would influence both the soil ecosystem and rhizosphere communities.

Thus, plants could play important roles in the shaping of soil and in the regulation of microbiomes growing in the rhizosphere, changing both the composition and amount of root exudates and recruiting specific soil microbiomes [9,10]. These modifications could influence the mobility and availability of MPs and NPs in the environment.

Curiously, damages induced by the presence of MPs and NPs in higher plants have been thoroughly investigated, but to our knowledge, no approach based on genetic engineering or synthetic biology has yet been suggested yet to overcome or mitigate plastic toxicity effects in plants.

Transcriptomics revealed the regulation of a large network of genes activated by the presence of MPs and NPs [38,39,43,44,45]. These catalogues of genes represent a powerful source to identify candidates able to improve plant tolerance to MPs and NPs.

From a physiological point of view, plants similarly respond to abiotic stress and plastic exposition by increasing the activities of ROS-detoxing enzymes, manipulating the expression of genes involved in nutrient uptake, as well as trace elements, and increasing the expression of chaperones and proteins able to guarantee the correct folding [48,49,50].

## 5. MPs and NPs Interaction with Trace Metals and Metalloids in Plants

Similar to other abiotic constraints, the contamination of soil by trace metals (TMs) and metalloids is a pressing issue for guaranteeing food safety, considering the dangerous effects of these elements on different aspects of plant physiology [48,51,52]. Plastic is an important factor regulating biological toxicity and the migration–transformation abilities of TMs in soil [53]. In particularly, MPs modify the abiotic properties of soil (namely pH value and cation exchange capacity), influence the chemical forms and bioavailability of nutrients and metals, impacting microbial, animal, and plant metabolism and their interactions [13,54]. The presence of MPs and NPs alters the bioavailability and content of TMs and nutrients [55]. Specifically, the effects of MPs and NPs can be contrasting, depending on the sizes, chemical types, and quantities, thus influencing the uptake of TMs and consequently affecting the plant physiology (Table 2).

Pb and Cd accumulated on PET–MPs particles; these particles are able to adsorb these metals both in single-element and mixed solutions. In particular, the rate of Pb adsorption on PET particles was lower than that of Cd. The adsorbed TMs were then desorbed in the artificial rhizosphere zone of *Triticum aestivum*, demonstrating that PET particles can act as a carrier, increasing the uptake of TMs in the rhizosphere [57].

In *Brassica napus*, PE–MPs increased the bioaccumulation of Cu^2+^ and Pb^2+^ [58]. The concomitant presence of these metals and PS–MPs increased the oxidative damage with respect to that reported by the single presence of Cu^2+^ and Pb^2+^ in terms of malondialdehyde content and higher SOD and POD activities [58]. Likewise, combinatory treatments induced detrimental effects also for turnip quality, showing lower content in soluble sugars and vitamin C [58]

Similarly, PS–MPs and PS–NPs improved the uptake of Cu, Pb, and Cd in lettuce. In particular, plastic particles increase the abundance of bacteria (e.g., *Clostridium* sp.) able to improve the bioavailability of TMs. Both MPs and NPs increased the content of Cu, Zn, Pb, and Cd. In particular, PS-NPs (at 1000 mg kg^−1^ of soil) augmented the levels of these elements in plants up to 52.6, 174, 10.3, and 33.2 mg kg^−1^, respectively [57]. Metabolomic analyses also showed that the simultaneous treatment with PS-NPs and TMs regulates different aspects of nutrient uptake, modifying the metabolism of ATP-binding cassette transporters and plant hormone signal transduction, therefore explaining the enhanced uptake in lettuce [59].

On the other hand, PSMPs and PSNPs can alleviate the toxic effects of Cd in *Brassica chinensis* L. and *Zea mays* L., respectively [25,60]. In contrast, the presence of different concentrations of PS–MPs in soil can affect different physiological parameters, such as biomass, photosynthesis, stomatal conductance, transpiration, and chlorophyll content, reducing the toxic effects of Cd. PS–MPs affect both Cd uptake and storage. In fact, plants exposed to Cd showed a concentration of 46.55 and 89.22 mg kg^−1^ in shoots and roots, respectively, while plants also treated with PS–MPs showed Cd content of about 31.30 and 75.98 mg kg^−1^ [25].

MPs and NPs are related to As and Pb uptake by plants [56,61,62]. The presence of PS–MPs and PTFE–MPs (polytetrafluorethylene) modify the bioavailability of arsenic (V) and arsenic (III) in soil [56]. The simultaneous presence of MPs and arsenic reduced the abundance of *Proteobacteria*, improving the occurrence of *Chloroflexi* and *Acidobacteria*; this outcome caused a reduction in the presence in soil urease, acid phosphatase, protease, dehydrogenase, and peroxidase, finally affecting the micro-environment of the rice rhizosphere. Therefore, PS–MPs and PTFE–MPs alleviate the effects of arsenic toxicity in soil, microbiomes, and plants [56].

Single or combined exposure of PS–NPs and polymers (such as polymethyl methacrylate nanoplastics, PMMA–NPs) showed the ability to modify different aspects of plant nutrition in *L. minor* L. [62]. Both single and combined treatments reduced the accumulation of micro- and macronutrients, namely N, P, K, Ca, Mg, and Mn. The effects of PS–NPs and PMMA–NPs indicated that the single-type particle treatment partially reduced As levels in plant tissues, but the combined treatment increased As uptake. Furthermore, As increased the content of PS–NPs and PMMA–NPs in *L. minor* L. [62]. Interestingly, Mamathaxim et al. [63] also showed comparable behavior in rice plants exposed to combined treatments with PS–MPs, PS–NPs, and As (III and V). No differences were reported comparing the uptake of As in the absence and presence of MPs. On the other hand, the presence of PS–NPs increased the presence of As (III) and As (V) in rice roots [63].

Plastics act as an effective abiotic stress, modifying the levels of different phytohormones, increasing the presence of abscisic acid (ABA), and decreasing the presence salicylic acid (SA) and jasmoic acid (JA) [62]. Interestingly, ABA showed the highest level in the presence of As, PS–NPs and PMMA–NPs, whereas under similar conditions, SA and JA showed the lowest levels [62].

The effects of NPs on phytohormone content reveal a possible role of these particles on plant signaling. The regulation of plant hormones affects gene expression (ABA or JA), contributing to the increase in chlorophyll biosynthesis (cytokinin) or optimizing the allocation of nitrogen (SA) [62]. Similarly, the effects on phytohormone regulation in the presence of PS were reported in rice [38] and tomato [64], but the mechanism regulating the role of NPs in this process has not yet been determined [65].

## 6. Crops Affected by MPs and NPs: A Threat for the Food Chain

Many studies have investigated the impact of ingestion of MPs and NPs on humans. These studies have shown a risk associated with plastic accumulation in human tissues, organs, and fluids. Their persistence induces specific biological responses and primarily affects the gastrointestinal tract and reproductive system. The accumulation of plastics in the organism can induce inflammation and immune and neurotoxic diseases [66,67,68]. In particularly, recent studies in human intestinal cells inoculated with fluorescent microspheres of PS–MPs showed significant accumulation of these particles in epithelial cells, with low cytotoxic effects but significant increases in ROS levels. These effects altered the correct cell morphology and function [68,69].

MPs and NPs are able to contaminate seeds, roots, culms, leaves, and fruits, depending on their size and type, including crops [70,71]. Considering this critical issue, different authors have investigated the effects of MPs and NPs in plants for human consumption and their effects on human health. Contamination by plastics negatively affects yields (Table 3), reducing crop productivity [72,73]. Examples include rice [74], spring onion [75], cucumber [76], and pea, which particularly showed a 30% yield reduction [75].

A further—and even worse—criticism is represented by the detection of MPs and NPs in edible fruits [73]. The presence and incidence of MPs and NPs in different vegetables and fruits purchased in both local and large distribution markets have been demonstrated [71]. Authors found MP and NP contamination in different edible species, namely two varieties of *Malus domestica* L., *Pyrus communis* L., and four vegetables: *B. oleracea* L., *Lactuca sativa* L., *Daucus carota* L., and *S. tuberosum* L.

Fruits accumulated up to 223,000 particles/gram (median value) in *M. domestica* and 176,500 particles/gram in *Pyrus communis* L. Conversely, *L. sativa* L. showed lower contamination, 52,050 particles g^−1^. The sizes of MPs and NPs retrieved in these crops ranged from 1.4 to 3.2 μm [71], with higher levels of MPs and NPs in fruits. Therefore, it could be concluded that NP contamination affects vegetable quality.

Foliar application of PS–NPs reduces the nutritional quality of *L. sativa* L. [78]. Significant reductions in total soluble proteins, total soluble sugars, and amino acids were observed in lettuce exposed to 1 mg L^−1^ of PS–NPs, together with changes in macro- and micronutrients [78]. The toxic effects of MPs and NPs were also observed in grains of *Oryza sativa* L. (*Poaceae*) and *Arachis hypogaea* L. (*Fabaceae*) [79]. In particular, peanut grains (developed underground) were more vulnerable to NPs. Both species showed the presence of PS–NPs in grain when cultivated in soil containing 250 mg kg^−1^ NPs, showing detrimental effects on grain quality. The presence of PS–NPs decreased peanut seed weight by 3.45%, while the rice empty-shell seeds increased by 35.45%. Furthermore, the presence of PS–NPs caused reductions in Ca, Mn, and Zn concentrations and in the levels of Tyr, Phe, Lys, and Arg. PS–NPs showed different effects on fatty acid content in rice and peanuts. MPs and NPs reduced the content of unsaturated fatty acid (UFAs) in rice grains, including C20:1 and C20:2, and saturated FAs (C12:0 and C17:0). Differently, exposure to 250 mg kg^−1^ PS–NPs in peanuts significantly reduced the levels of UFAs (C16:1, C18:3), but different SFAs (C10:0, C15:0, and C16:0) increased [79].

## 7. Plastic in the Environment: Research Gaps and Possible Strategies to Overcome Plastic Stress

Most of current knowledge about plastic pollution in the environment has been focused on aquatic ecosystems [2]. To limit damage, the utilization has been proposed of eco-sustainable and easily perishable materials in the marine environment. Since 2018, scientists have become increasingly interested in evaluating how plastic particles affect soil agricultural environments and crops. Therefore, this theme is relatively young.

The generation of plastic waste and the effects of these particles are pressing challenges in both industrial and developing countries, where they are related to a greater urbanization and higher economic growth [1].

The effects of globalization have shifted the environmental impacts to other countries around the world. Approximately 5–20% of imported plastic waste in emerging economies has no market value; therefore, this waste ends up in the environment or in open dumps or is burned [1]. As shown in this manuscript and the literature, plant exposure to MPs and NPs leads to toxicity symptoms, including growth inhibition, alterations in mineral homeostasis and photosynthesis efficiency, decreased cell division, and genotoxic effects [4,22,38,43]. These effects could vary as a function of plant species and therefore will possibly involve modifications to the composition of plant communities and primary production [12].

A number of questions and solutions about the toxic effects of MPs and NPs on the environment and plants remain unanswered. One of the most important questions is how plastic particles can gain entry into plants, How can plants activate a defense by closing off these particles? [87]. The identification of specific weights, sizes, types, and directions of MPs and NPs at different levels, such as in specific plant species, communities, rhizosphere communities, and whole ecosystems, will be challenging [12]. Similar to other abiotic stresses (drought, salinity, etc.), the identification of those genotypes (local or improved) able to tolerate or avoid the effects of plastic toxicity will be also critical. The genetic and phenotypic peculiarities must be investigated in their ability to avoid MP and NP uptake [12,87] or to retrieve the abilities of genotypes tolerant to typical abiotic constraints.

Even though the effects of plastics on the terrestrial ecosystem have been recently documented, there have still been few studies exploring methods of plastic removal or degradation. In water environments, the removal of MPs and NPs is a difficult process considering their small size [88]. MPs remain stable during physical processes, such as coagulation, sedimentation, and screening, due their high polymeric flotation properties [89]. Other strategies involve different environmental factors that affect the physical structures of plastics or soil micro- and meso-fauna (microorganisms and invertebrates) [90,91]. In particular, the use of biotic processes would be an interesting alternative to overcome the presence of plastics in the soil environment and improve plants’ tolerance of MPs and NPs. However, the consumption of plastic by organisms and their degradation by host gut enzymes, related to their microbiome, require further confirmation [92].

## 8. Conclusions

The results and indications reported in this review summarize various aspects of MPs’ and NPs’ effects on agriculture soils and plant physiology. Currently, MPs and NPs, composed of different kinds of plastics, (polystyrene, polyvinyl chloride, polyethylene, polypropylene), impact various plant processes, including photosynthesis, nutrient uptake, growth, and development, inducing osmotic and oxidative damages similar to those reported with abiotic perturbations. The effects of plastics can be contrasting, depending on their dimensions, compositions, and concentrations. Examples are the relationships with trace metals, root morphology, or activation of the antioxidant systems. In fact, a beneficial impact of plastics has been reported in counteracting the detrimental effects of trace metals, but—generally—the presence of MPs and NPs in soil (or in hydroponics) induced the activation of typical stress response pathways, including oxidative burst and protein damage. Scavenging enzymes also differently respond to MPs and NPs. An emblematic case is catalase, showing a short-term activity increase in *Oryza sativa* L. and *Lemna minor* L., a long-term activity increase in *Solanun lycopersicum* L., and a long-term activity decrease in *Vicia faba* L. These results highlight the complexity and peculiarity of this specific anthropic stress.

Furthermore, MPs and NPs are able to regulate the expression of thousands of genes perturbing different metabolic pathways in important crops, such as rice, maize, and tobacco, similar to other abiotic stresses. This transcriptomic reorganization particularly influences biological processes, such as protein processing, biosynthesis of fatty acids and secondary metabolites, or the hormonal transduction pathways (ABA, SA, JA, etc.).

Finally, crop productivity and quality are seriously endangered by MPs and NPs. This endangerment will be especially true for agricultural production, considering the extensive and essential use of plastics. Consequently, in coming years, this issue could have critical effects on food demand, food quality, and human health. The “plastic problem” needs to be brought into the spotlight because it represents a critical challenge for plant and environmental scientists in the years to come.

## Figures and Tables

**Figure 1 plants-12-03717-f001:**
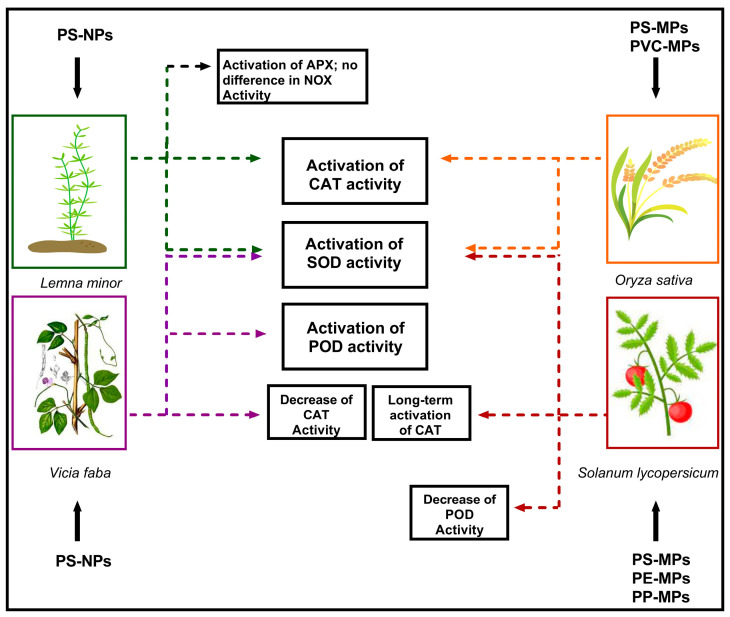
Scheme of the activation of scavenging enzymes SOD (superoxide dismutase), CAT (catalase), APX (ascorbate peroxidase), POD (peroxidase), and NOX (NADPH-oxidase) in different plant species exposed to MPs and NPs. Colors indicate different plant species.

**Table 1 plants-12-03717-t001:** Estimation of plastic consumption for agricultural practices.

Year	Location	Plastic Estimation	Reference
2012	Worldwide	6.5 million tons/year	[14]
2014	China	1.4 million tons year *	[15]
2015	Worldwide	7 to 9 million tons/year	[16]
2019	EU	708.000 tons/year **	[1]

*= only for mulching; ** = not including plastic packaging for agricultural products.

**Table 2 plants-12-03717-t002:** Effects of MPs and NPs on trace metal uptake.

Plant Species	Cultivation	Plastics Type	Concentration	Trace Metal (TM)	TM Concentration	Effects on HMs Uptake	Reference
*Brassica chinensis*	Soil	Polystyrene–MPs	0.5–2%	Cd	10 mg kg^−1^	Decreased uptake	[25]
*Oryza sativa*	Soil	Polystyrene–MPs	0.25%	As	1.4, 24.7 and 86.3 mg kg^−1^	Decreased bioavailability	[56]
*Oryza sativa*	Soil	Polytetrafluorethylene–MPs	0.5%	As	1.4, 24.7 and 86.3 mg kg^−1^	Decreased bioavailability	[56]
*Triticum aestivum*	Soil	Polyethylene terephthalate–MPs	1 g in 20 mL	Pb	0.31 to 100.84 μg L^−1^	Accumulation on PET–MPs	[57]
*Triticum aestivum*	Soil	Polyethylene terephthalate–MPs	1 g in 20 mL	Cd	5020.65 to 599,436.62 μg L^−1^	Accumulation on PET–MPs	[57]
*Brassica napus*	Soil	Polyethylene–MPs	0.01 to 0.1% of soil weight	Pb	25–50 mg kg^−1^	Increased bioaccumulation	[58]
*Brassica napus*	Soil	Polyethylene–MPs	0.01 to 0.1% of soil weight	Cu	50–100 mg kg^−1^	Increased bioaccumulation	[58]
*Lactuca sativa*	Soil	Polystyrene–MPs	100 and 1000 ng kg^−1^	Cu, Pb, Cd	82, 174.84, 42.0, 0.20 mg kg^−1^	Increased uptake	[59]
*Lactuca sativa*	Soil	Polystyrene–NPs	100 and 1000 ng kg^−1^	Cu, Pb, Cd	82, 174.84, 42.0, 0.20 mg kg^−1^	Increased uptake	[59]
*Zea mays*	Hydroponics	Polystyrene–NPs	10 and 100 mg L^−1^	Cd	2 mg/L and 10 mg L^−1^	Antagonistic effects	[60]
*Taraxacum asiaticum*	Hydroponics	Polystyrene–MPs	1, 5 and 10 mg L^−1^	Pb	10 mg L^−1^	Increased uptake	[61]
*Lemna minor*	Hydroponics	Polystyrene–NPs	100 mg L^−1^	As	100 μM	Decreased uptake	[62]
*Lemna minor*	Hydroponics	Polymethyl methacrylate–NPs	100 mg L^−1^	As	100 μM	Decreased uptake	[62]
*Lemne minor*	Hydroponics	Combined–NPs	100 mg L^−1^	As	100 μM	Decrease uptake	[62]
*Oryza sativa*	Hydroponics	Polystyrene–MPs	50 mg L^−1^	As	250 μg L^−1^	No effects	[63]
*Oryza sativa*	Hydroponics	Polystyrene–NPs	50 mg L^−1^	As	250 μg L^−1^	Promote accumulation	[63]

**Table 3 plants-12-03717-t003:** Effects of MPs and NPs on crop productivity. Abbreviations: primary polyamide (PA); polyester (PES); polyethylene high density (PEHD); polypropylene (PP); polystyrene (PS); polyethylene terephthalate (PET); polyvinyl chloride (PVC); polyethylene (PE); biodegradable microplastic (Bio–MPs); low-density polyethylene (LDPE–MPs).

Crops	Plastics Type	Effects	Yield	Reference
*Cucumis sativus*	PS	Reduction: soluble protein and sugar, vitamin C, and mineral element content; Increment: total soluble protein	50% reduction in vitamin C and sugar content	[76]
*Pisum sativum*	MP	Reduction: pods and beans for pod numbers; Increment: shoot growth Changes: amino acid profile	20.5% reduction in bean per pod number	[77]
*Lactuca sativa*	PS	Reduction: dry weight, height, leaf area, pigment content, shoot-to-root dry biomass ratios, macro- and micronutrient and amino acid content, photosynthetic performance, and chlorophyll and carotenoid content	Dry weight −27.3%; height −27.3%; leaf area −19.2%	[78]
*Oryza sativa*	PS	Reduction: seed weight, nutritional quality	Seed weight −3.45%	[79]
*Arachis hypogaea*	PS	Reduction: weight and total grain number per plant, nutritional quality	Empty shell number x plant +35.45% Seed-setting rate −3.02%,	[79]
*Solanum lycopersicum*	PET; PVC	Reduction: shoot growth, photosynthesis, fruit development and quality	No. of fruits x plant −28%; fruits’ fresh weight −25%	[80]
*Cucurbita pepo*	PP; PE; PVC; PET	Reduction: shoot growth, leaf size, chlorophyll content; photosynthetic efficiency	Fresh weight of shoots −35%	[81]
*Triticum aestivum*	PS	Reduction: macro- and micronutrient content Increment: biomass and root elongation Changes: leaf metabolic profiles	Shoot biomass +87.1%; root biomass +116.5%, shoot root ratio −27.3%	[82]
*Daucus carota*	PS	Reduction: biomass; soluble protein, vitamin C, soluble sugar, α-carotene, and β-carotene and chlorophyll content	Root biomass −12.7%; leaf biomass −21,4%	[83]
*Phaseolus vulgaris*	LDPE	Reduction: chlorophyll content Increment: leaf area, root length, specific root nodules	No significant data obtained for fruit biomass	[84]
*Phaseolus vulgaris*	Bio–MPs	Reduction: root, shoot, and fruit biomass, chlorophyll content, leaf area. Increment: root length	Fruit biomass −43%	[84]
*Solanum lycopersicum*	PET; PVC	Reduction: plant number, fruit production Increment: shoot and root biomass	No. of plants −25%; no mature tomatoes; shoot biomass increase +2.21 fold change; root biomass +2.89 fold change	[85]
*Zea mays*	PS	Reduction: biomass root and number of lateral roots	Root dry weight −49.4%	[86]
*Lactuca sativa*	PS	Reduction: root lengths and biomass, germination	Germination index −36.0%; root dry weight −50%	[86]

## Data Availability

Not applicable.

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
