# Peer review of "Plastic in the Environment: A Modern Type of Abiotic Stress for Plant Physiology"

_plants, 2023, doi:10.3390/plants12213717_

Round 1

Reviewer 1 Report

Comments and Suggestions for Authors

Title: Plastic in the environment: a modern type of abiotic stress for 3 plant physiology

The aim of this work is to review the current knowledge and findings about occurrences, effects, accumulation and influences of MPs and NPs on different plant organisms.  Particularly, the detrimental effects triggered by MPs and NPs will be summarized including nutrient uptake, photosynthetic injuries, oxidative stress, ROS production, transcriptomic modification and protein damages. The relationships between plastics and metals in the soil, how MPs can act as carriers of these pollutants, and the consequences on plant physiology will be discussed.

The title and subject of the manuscript are very interestin. The analysis of the published data was provided with a sufficient level of scientific novelty. However, some corrections should be considered as listed below:

Please provide “Keywords” after Abstract section. Keywords should appear in alphabetical order, and do not repeat words in the Keywords that have been previously cited in the Title.

The authors should mention about "research gaps", the research gaps are not clear in the whole manuscript. It is better to provide a paragraph above the conclusion section to clearly describe what the research gaps are.

The English language should be polished.

Line 133-134, Please add the below references at the end of sentence:

These results suggest the activation of different abiotic stress response pathways to oxidative and osmotic damages, respectively (Mei et al., 2023; Sachdev et al., 2021).

Mei, X., Dai, T. & Shen, Y. Adaptive strategy of Nitraria sibirica to transient salt, alkali and osmotic stresses via the alteration of Na+/K+ fluxes around root tips. J. For. Res. 34, 425–432 (2023). https://doi.org/10.1007/s11676-022-01486-1.

Sachdev S, Ansari SA, Ansari MI, Fujita M, Hasanuzzaman M. Abiotic Stress and Reactive Oxygen Species: Generation, Signaling, and Defense Mechanisms. Antioxidants (Basel). 2021 Feb 11;10(2):277. doi: 10.3390/antiox10020277.

There are many references that are relatively old.Although authors included many references, they missed recent references.

There are some imprecisions in the interpretation of literature, please check them.

In summary, authors can improve the review by incorporated the most recent literature and pointing out in detail the new research routes to better understand the interactions of plastics and plants.

Author Response

See  file

Reviewer 2 Report

Comments and Suggestions for Authors

The manuscript is very important topic and it original work. It contains new science and current knowledge and it include sufficient data. It's worth noting that as plastic pollution continues to increase globally, its effects on plants are an area of growing concern for ecologists, environmental scientists, and conservationists. Mitigating plastic pollution and finding ways to remediate its effects on plant ecosystems is an important aspect of environmental stewardship and conservation efforts.

I think that the topic content of this manuscript is also suitable for the international audiences of Plants. However it need minor revision before it ready for publication. I encourage the authors to rework the manuscript. Here are some suggestions :

 1-      While plastic pollution poses significant challenges to plant physiology and ecosystem health, there are numerous limitations and complexities in understanding and mitigating these effects. The manuscript lack a section to show into some of these limitations and complexities (long term effects, interaction with other stress, ecosystem-level effects…).

 2-      The use of the term "heavy metals", which was never internationally defined, should be discouraged. The term "heavy metals" should be replaced by "trace elements" (TE), "metals" "trace metals", "toxic elements" (for metal whose toxicity is proven) or "potentially toxic elements".

 3-      The manuscript is need to improving in English language, please improve it.

Comments on the Quality of English Language

The manuscript is need to improving in English language, please improve it.

Author Response

See file

Reviewer 3 Report

Comments and Suggestions for Authors

Review article “Plastic in the environment …… plant physiology” present literature on a new type of plant stress i.e. plastic. They cover impacts of micro- and nano-plastic particles on plants in the terrestrial environment. They also overviewed the detrimental effects of above said particles in plants especially crops. My suggestion is:–

Include a table for section 6 (crops affected by…..) as per literature available. In brief, How many/type of crops get affected, what was the effect, effect on physiology/metabolism, damage estimated, yield loss and references etcetera.

Include/discuss following as different section

1.      Possible technologies/ or practices to overcome this plastic stress.

2.      How plant reacts with plastic stress and how developed resistant naturally?

3.      Possibility or role of genetic engineering/molecular approaches to overcome plastic stress?

4.      Role of metagenome/plant rhizosphere in the context of plastic stress.

5.      Effect of plastic on crop/plant root architecture

6.      Plant Signaling in the context of plastic stress

7.      Plastic uptake and water potential relation

8.      Connection between plastic and physiology

Author Response

See file

Reviewer 4 Report

Comments and Suggestions for Authors

This review outlines the physiological changes in plants caused by MP & NPs.

Major Concern:

In Line 119, it is stated that NPs can directly penetrate plant cell walls, whereas MPs cannot. Given this, the impact of these two sizes of plastics on plant development may differ. There seems to be a subtle difference in the POD response to these two sizes of plastics, as shown in Figure 1. Moreover, differences in dosage were observed in the studies that reported ROS changes (Lines 152-184) and in the summaries of heavy metal uptake (Table 2) between MPs and NPs. In the context of a review, it is crucial to approach this critically. I recommend either distinguishing between MPs and NPs as two separate abiotic stressors or including additional peer-reviewed references to support the notion that both MPs and NPs can be classified under the same category of abiotic stress.

Minor Suggestions:

Figure 1 is not clear enough. It's challenging for readers to discern the specific response from each study. For instance, it's unclear whether V. faba or L. minor exhibited an activated POD response in the presence of PS-NPs.

On Line 100, the abbreviation "PP" is used, and on Line 123, "PS" is used. Neither of these abbreviations has been defined. Please provide clarity on these terms.

Author Response

See file

Round 2

Reviewer 3 Report

Comments and Suggestions for Authors

Revision is satisfactory

Reviewer 4 Report

Comments and Suggestions for Authors

Authors have responded comments well.